# Body-Centered Double-Square Split-Ring Enclosed Nested Meander-Line-Shaped Metamaterial-Loaded Microstrip-Based Resonator for Sensing Applications

**DOI:** 10.3390/ma15186186

**Published:** 2022-09-06

**Authors:** Air Mohammad Siddiky, Mohammad Rashed Iqbal Faruque, Mohammad Tariqul Islam, Sabirin Abdullah, Mayeen Uddin Khandaker, Nissren Tamam, Abdelmoneim Sulieman

**Affiliations:** 1Space Science Centre (ANGKASA), Institute of Climate Change (IPI), Universiti Kebangsaan Malaysia, Bangi 43600, Malaysia; 2Department of Electrical, Electronic & Systems Engineering, Universiti Kebangsaan Malaysia, Bangi 43600, Malaysia; 3Centre for Applied Physics and Radiation Technologies, School of Engineering and Technology, Sunway University, Bandar Sunway 47500, Malaysia; 4Department of General Educational Development, Faculty of Science & Information Technology, Daffodil International University, DIU Road, Dhaka 1341, Bangladesh; 5Department of Physics, College of Sciences, Princess Nourah Bint Abdulrahman University, P.O. Box 84428, Riyadh 11671, Saudi Arabia; 6Department of Radiology and Medical Imaging, Prince Sattam Bin Abdul Aziz University, Alkharj 16278, Saudi Arabia

**Keywords:** double negative, metamaterial, microstrip resonator, sensing

## Abstract

The strong localization of the electric and magnetic fields in metamaterial-based structures has attracted a new era of radiation fields in the microwave range. In this research work, we represent a double split ring enclosed nested meander-line-shaped metamaterial resonator with a high effective medium ratio layered on a dielectric substrate to enhance the sensitivity for the material characterization. Tailoring a metallic design and periodical arrangement of the split ring resonator in a subwavelength range introduced field enhancement and strong localization of the electromagnetic field. The design methodology is carried out through the optimization technique with different geometric configurations to increase the compactness of the design. The CST microwave studio is utilized for the extraction of the scattering computational value within the defined boundary condition. The effective parameters from the reflection and transmission coefficient are taken into account to observe the radiation characteristics for the interaction with the applied electromagnetic spectrum. The proposed metamaterial-based sensor exhibits high sensitivity for different dielectric materials with low permittivity values. The numerical data of the frequency deviation for the different dielectric constants have shown good agreement using the linear regression analysis where the sensitivity is R^2^ = 0.9894 and the figure of merit is R^2^ = 0.9978.

## 1. Introduction

Microwave sensor [1,2] applications in the domain of liquid or solid materials have been created by several scientists and research organizations due to the high localization of electromagnetic fields. Microwave sensors based on MTM offer several benefits and drawbacks. While certain sensors, such as those used in microfluidic sensor applications, have a high-quality factor, others may have cheap costs and great sensitivity. The complex dielectric constant is used as an input in metamaterial-based [3,4,5] sensor applications since it directly impacts the variation in the resonance frequency. The electromagnetic properties through the metal-dielectric-based structure in the subwavelength dimension exhibit complex responses for the tailored metallic design and geometric orientation, which is not shown in nature. The exotic properties of the metamaterial unit cell cover different fields [6,7,8,9,10,11,12,13,14,15,16,17] for wireless communication systems. The inductive and capacitive responses are responsible for the resonant properties of the applied electromagnetic properties, and artificial magnetic resonance introduces strong coupling into the metamaterial unit cell. The negative effective parameters have enhanced the confinement of electromagnetic fields in the substrate. The compactness of the design can be achieved by the split ring technology and the design optimization of the metallic layer on the dielectric substrate.

The detection of chemical samples with a tight dielectric response is difficult since the detection concept is based on fluctuations in the dielectric characteristics. Carbon steel fiber ratios of 0.5 percent, 1 percent, and 1.5 percent were created, and several metamaterial designs were constructed in this study to propose a sensor to detect the crack in the concrete sensor where the resonance frequency changes, and temperature and humidity treatment techniques were also used. The suggested design may be effectively used in material sensor applications and physical parameter observations such as humidity treatment and temperature sensing, according to simulated metamaterial-based sensors with experimental research findings [18]. A complementary circular spiral resonator (CCSR) was focused on a tiny volume of the sample on the substrate. This design was used to detect a variety of electromagnetic characteristics for different materials under test (MUT). The suggested sensor exhibited effective electric permittivity, and magnetic permeability was calculated using scattering characteristics where the permittivity of MUT was varied to perform a sensitivity study [19]. The terahertz (THz) sensor was developed, which incorporated a split-ring resonator with four gaps, and a nested-type square-ring resonator. When the surface permittivity of the metamaterial (MTM) structure changes, the two resonant components form matching resonance on the transmission spectrum curve within 0.1 to 1.9 THz, and each of these resonant valleys displays distinct redshifts [20]. The effort involves the design and fabrication of a novel metamaterial-based sensor that can be used to detect liquid substances in the frequency range of 8 to 12 GHz, where the resonance frequency showed the shift of 250 MHz, 200 MHz, 250 MHz, 150 MHz, and 50 MHz for the samples, which made this conceivable [21]. The author suggested a new metamaterial absorber-based sensor in this research to determine liquid molecules based on electrical properties. Gammadion resonators used a copper metallic layer on top of an FR4 substrate and a 2 mm air gap between the copper ground plate and the backside of the resonator for adjusting resonance frequency. The relative dielectric constants of chemical liquids were measured in the related frequency range where the absorption value of the sensor structure was analyzed in the air gap region for each chemical sample [22]. This work described a metamaterial sensor that uses a double-split, complementary, square-ring resonator (DS-CSRR) for dielectric material measurement, particularly coal powder. The design was tailored for the optimum deep notch depth transmission coefficient performance (Magnitude of *S*_21_). A transmission coefficient sensitivity study was performed with regard to structural dimensions. The S parameters of this sensor were used to derive metamaterial characteristics of double-negative permittivity and permeability [23]. For refractive index sensing, a new cavity design based on a racetrack-integrated circular cavity built on a metal-insulator-metal (MIM) waveguide was proposed. The optimized cavity design with small variations in the geometric layer exhibited a remarkable shift in the sensor device [24]. A split-ring resonator-based microwave sensor was demonstrated for the detection of different concentrations of the liquid samples using complex permittivity characterization [25]. A semi-circle-shaped, asymmetric, split-ring resonator was designed for sensing the solution of glucose in the microliter range [26]. A modified split-ring resonator was developed for electric field sensing, showing strong localization of the electromagnetic field. This SRR resonator-based sensor is designed with three concentric pairs of rings for industrial, scientific, and medical band applications [27]. A theoretical approach for analytically solving wave propagation through two-dimensional (2D) inhomogeneous slab waveguides was introduced using the effect of a refractive index-guided mode profile [28].

In this research work, the current effort involves the design of a novel metamaterial-based sensor that can be used to detect substances within the frequency range of 4 to 8 GHz. In order to maximize the desired performance and compactness of the resonator, the square-ring, enclosed, nested, meander-line-shaped metamaterial unit cell is implemented in the center of the proposed metamaterial-based unit cell. The computational findings demonstrated that the suggested sensor with body-centered metamaterial exhibits strong localization of the electromagnetic magnetic field, enhances the compactness of the structure, and increases the capability of detecting materials for different dielectric constants. Moreover, the steadily observed findings reveal that the suggested sensor, which operates in the C-band at approximately 6.2 GHz, can distinguish various MUT based on their distinct resonance frequency shift and shows the good linear correlation in the sensitivity analysis for low permittivity values.

## 2. Metamaterial Incorporated Sensor Design and Method

### 2.1. Metamaterial Unit Cell Design 

For the proposed metamaterial unit cell, shown in Figure 1, the design is composed of two layers where copper is utilized as a top metallic layer, and the dimensional data specifications are given in Table 1. The thickness of the copper layer is 0.035 mm, and the Rogers RT 6002 substrate is used as the bottom layer of the substrate. The thickness of the dielectric substrate is 1.524 mm. The metallic layer of the metamaterial unit cell provides the electric resonance response, and the substrate layer establishes the polarization state to produce the maximum displacement of the bound charges. In the proposed metamaterial unit cell, the dimension is specified for a length of 9 mm, a width of 9 mm, and a thickness of 1.524 mm. Dual split rings are deployed on the dielectric substrate where the split between the metal is 0.5 mm. The outer metallic split ring has a width of 7.9 to 8.4 mm, and the width of the middle split ring is 6.3 to 6.8 mm, where the gap between the two split rings is 0.3 mm. The width of the diagonally bent metallic slot is 1.2 mm, where two triangular-shaped metallic arms are connected to complete the metallic split ring and the length of the bent metallic arm is 2.8 mm. The split between the metallic layer in the different coordinate systems produces the capacitance effect with the inductance of the metallic strip for the proposed metamaterial unit cell.

### 2.2. Boundary Condition

The boundary condition for this proposed metamaterial unit cell is defined with an effective dimension to extract the scattering properties of the applied transverse electromagnetic wave. In Figure 2, the transverse electromagnetic signal travels in the free space along the Z-direction through the metamaterial unit cell. The electric field and magnetic field are specified in the X- and Y-directions, respectively. The field coefficient for the applied EM wave in different arrangements of the metallic layer and split between the metallic strip produce the new scattering properties, and the polarization takes place in the transverse plane. The periodic distribution of the suggested metamaterial properties is observed with a specified physical topology to avoid periodic artifacts in the numerical simulation. A tetrahedral mesh is adopted with a finite integration technique to provide high-quality meshing facilities for the necessary sampling of the complex structure.

The reflection coefficient and transmission coefficient are used to calculate the effective parameters, such as negative permittivity, negative permeability, and the negative refractive index. The impedance discontinuity is responsible for the reflection of the wave and the propagation through the medium relative to the transmission at a certain frequency. The complex ratio of the reflected and transmitted wave to the incident wave in the higher-frequency region is referred to as the reflection coefficient and transmission coefficient. The permittivity of a substrate depends on the polarizability of the electric dipole in the material for an incident time-varying electric field and permeability related to the magnetic moment of the particle with an external time-varying magnetic field. After the interaction with the metal-dielectric, medium EM properties in the higher-frequency region depend on the momentum of the bound charge and free electrons where the dimension of the metamaterial slab will be kept in the subwavelength dimension and maintain a certain distance from the source for effective interaction. The resonant frequencies of the transmission coefficient for this metamaterial unit cell are located at 2.21, 4.1, and 5.6 GHz with a bandwidth of (2.16–2.23), (3.88–4.25), and (5.32–5.82) GHz, and reflection coefficients at 2.36, 4.53, and 6.42 GHz with a bandwidth of (2.30–2.44), (4.45–4.61), and (6.23–6.67) GHz, which are shown in Figure 3.

### 2.3. Design Optimization

The wave properties of wave propagation in the medium are determined by the electromagnetic wave interference pattern. The interplay of the time-variable electric and magnetic fields, which display peculiar scattering behaviors for the continuous finite slab, is connected to the characterization of the metal-dielectric based resonator. The optimization process for the design selection has been carried out, which is shown in Figure 4, and the corresponding result of the transmission coefficient *(S*_11_*)* and reflection coefficient *(S*_21_*)* is shown in Figure 5a,b. Different layouts of the metallic strip in different shapes cause a change in the electric and magnetic resonance, where the resonant frequencies of *S*_11_ and *S*_21_ are shifted due to the mutual impedance effect of the metamaterial unit cell. The two resonant frequencies at the lower region are responsible for two split rings on the metamaterial design. In the proposed metamaterial unit cell, two square-shaped enclosed metamaterial unit cells with a metallic inner, which is diagonally connected to a metallic arm, exhibit resonant frequencies of *S*_21_ at 2.5, 4.9, and 10 GHz and *S*_11_ at 2.74, 6.0, and 11.25 GHz. The inductive values for the square-shaped inductor for the time-varying electromagnetic field are introduced with split-ring technology where the capacitive response for the split slot is responsible for producing the resonance frequency to the lower-frequency region, which increases the compactness of the design. The diagonally slotted inner metallic ring in design A2 shows the resonant frequencies of *S*_21_ at 2.4, 4.9, 10, and 10.5 GHz and *S*_11_ at 2.58, 6.11, and 11.82 GHz. Two triangular metallic arms coupled with shunt inductance in design A3 produce the resonant frequencies of *S*_21_ at 2.6, 5.0, and 10.3 GHz and *S*_11_ at 2.60, 5.84, and 10.84 GHz. In design A4, the diagonally bent, closed-loop inner metallic arm is introduced into the inductive loop with the parallel capacitance between the parallel split arms, and the dielectric material produces the resonant frequencies of *S*_21_ at 2.53, 4.9, 10.28, and 11.0 GHz and *S*_11_ at 2.60, 5.87, 11.55, and 12.5 GHz. In design A5, the inner metallic ring is diagonally slotted in two corners, which changes the mutual impedance response with a long inductive patch in the inner bent metallic ring. The slotted dimension with 0.5 mm is carried on for the inner metallic ring to introduce a new resonant frequency within the C band *S*_11_ at 2.4, 6.0, 11.4, and 12.3 GHz. The inner split ring is split into two parts, which introduce the capacitance effect for this metamaterial structure. In design A6, the inner nested meander-line-shaped metallic ring is connected and diagonally split at the upper side, which increases the inductance of the proposed structure. Due to an increase in the length of the metallic arm, which increases the inductive values in series for the proposed design, the resonant frequencies shift to the lower-frequency range, which enhances the compactness of the metamaterial unit cell for three resonant frequencies. The resonant frequencies are located at 2.36, 4.54, and 6.39 GHz for the reflection coefficient and 2.22, 4.16, and 5.63 GHz for the transmission coefficient.

### 2.4. Equivalent Circuit Model

Figure 6a depicts the equivalent circuit model for the proposed (SRR) metamaterial design, and the results are shown in Figure 6b,c. The metallic arms are responsible for the inductive response, which are represented as L1, L2, L3, L4, L5, L6, L7, L8, L9, L10, L11, and L12 in the equivalent circuit. Multiple wires and the junction of the metallic arm represent the equivalent inductances in the corresponding positions. The equivalent capacitances of the proposed design are C1, C2, C3, C4, C5, C6, C7, C8, and C9. The values of inductance, L1 and L2, are connected to the specific structure of the circuit element, therefore the construction of the metamaterial unit cell with a known material is derived from the values of the inductances. The capacitance is defined by the capacitive effects of the dielectric substrate and the impact of the samples, according to this design. The equivalent circuit is validated by ADS 2019 software. However, the corresponding results of the reflection and transmission coefficients show discrepancies with the simulated result of CST 2019 software due to the lumped element network and computing technique. The capacitive effects on both sides of the gaps are defined by the dielectric substrate, channels, and surrounding space of the sensor. The capacitive term refers to the contribution of the load’s liquid sample, where C is the capacitance of an empty channel and is the complex permittivity for the dielectric substrate. The dielectric materials around the gaps determine the value of the effective capacitance [29]. For this proposed sensor, the first outer-square split-ring with an inductance of L1 and L2 and the parallel LC branch is considered for the split of the metallic arm, which are split into two branches as (C1, L3) and (C2, L4). Consequently, (C4, L7) and (C5, L8) are the middle square split ring with the inductance of L5 and L6, and (C7, L9) and (C8, L10) are the inner nested meander-line-shaped metallic ring with the inductance of L11 and L12. Moreover, the capacitance (C3, C6, C9) is used to represent the capacitance between the metallic rings and dielectric substrate. L1 = 2 nH, L2 = 1.45 nH, L3 = 0.9 nH, L4 = 1.45 nH, L5 = 0.5 nH, L6 = 0.5 nH, L7 = 2.3 nH, L8 = 2.3 nH, L9 = 0.5 nH, L10 = 0.5 nH, L11 = 1.45 nH, L12 = 0.9 nH, and C1 = 2.3 pF, C2 = 1.13 pF, C3 = 1.5 pF, C4 = 1.5 pF, C5 = 1.5 pF, C6 = 1.2 pF, C7 = 1.1 pF, C8 = 0.5 pF, C9 = 0.5 pF. This subwavelength-based metamaterial-based resonator acts as an LC resonant circuit, and the resonant frequency can be written as
(1)$$f=\frac{1}{2\pi \sqrt{LC}}$$

For a single-loop metallic strip, the total inductance can be expressed as
(2)$$L_{\left(nH\right)}\;=1.257\;\times \;10^{-3}a\left[\ln\left(\frac{a}{w+t}\right)+0.078\right]K_{g}$$
where a is the radius of the metallic loop, *W* is the width of the metallic strip, *h* is the thickness of the dielectric substrate, and *t* is the thickness of the copper layer. For this proposed metamaterial unit cell, the metallic arms on the dielectric substrate are responsible for the inductive response, and the inductor values are represented in the equivalent circuit model where the approximate value using the inductive expression is utilized for the validation of the model.
(3)$$Kg_{\;}=0.57-0.145\;\ln\;\frac{W}{h}$$

The total capacitance for a single loop can be expressed as:(4)$$C\;(\mathrm{pF})=\frac{\varepsilon_{e}10^{-3}\;K\left(k\right)\;}{18\pi \;K^{\prime }\left(k\right)}\;(N-1)l$$
where *K* = $tan^{2}(\;\frac{a\pi }{4b}\;$) and *a* = $\frac{w}{2};\;b=\frac{w+s}{2}\;$.

W is the width of the conductor and S is the space between two parallel plates. $\varepsilon_{e}$ is the effective dielectric constant. The capacitance effect between the metal and the dielectric substrate, two parallel plates, and the fringing field is also taken into account, and the inductance value can be varied for different factors such as the length, turn ratio, and width of the metallic strip. The capacitive expression for this proposed metamaterial unit cell is represented by the capacitive effect in the dielectric substrate with the metallic arms on the design. The approximate capacitive values are considered for different geometric orientations of the metal with the dielectric substrate, which depends on the effective dielectric constant and dimensions of the substrate material.

### 2.5. Effective Parameter

The polarization effect of the metal-dielectric-based artificial subwavelength inclusion, the degree of magnetization of the conducting layer for various geometric orientations, and the propagation of the electromagnetic wave through the resonator are all addressed as effective parameters. The manufactured unit cell’s shape is arranged in a periodic pattern, and a metallic loop of a specific size has negative permittivity and permeability. The refractive index characterizes the propagation of the applied electromagnetic wave through the medium and is introduced by artificial magnetism qualities with negative permittivity in a higher-frequency zone. The Nicolson–Ross–Weir [30] method is used for effective parameters such as the effective permittivity, effective permeability, and refractive index from the transmission coefficient (*S*_21_) and the reflection coefficient (*S*_11_), and the results are shown in Figure 7. The effective parameters of the suggested metamaterial unit cell can be derived as follows:

Equation (5) is stated the effective permittivity ($\varepsilon_{r}$) can be expressed as:(5)$$\varepsilon_{r}=\frac{2}{j\pi fd}\times \frac{\left(1-S_{21}-S_{11}\right)}{\left(1+S_{21}+S_{11}\right)}$$

Equation (6) is stated the effective permeability (*μ_r_*) can be derived as:(6)$$\mu_{r}=\frac{2}{j\pi fd}\times \frac{\left(1-S_{21}+S_{11}\right)}{\left(1+S_{21}-S_{11}\right)}$$

### 2.6. Sensor Design

The metamaterial unit cell design in the subwavelength region, which is the basic component of the metamaterial-based sensor suggested in this study, is shown schematically in Figure 7. Because of its simplicity and sensitivity to the spectrum of the reflection coefficient changes, the metamaterial structure is selected as a fundamental building block of microstrip-based sensors. The Quasi-TEM wave is utilized to observe the scattering properties through the transmission of the design. Copper (annealed) has an electrical conductivity of 5.96 × 10^7^ s/m, whereas the substrate is Rogers RT 6002 (lossy) with a dielectric constant of 2.94. The following are the dimensions of the SRR construction depicted in Figure 7b,c: The patch width (W) is 12 mm, the patch length (L) is 12.5 mm, the line width (B) is 0.5 mm, the side width (A) is 5.0 mm, the dimension of the metamaterial unit cell (D) is 9 mm, the length of the transmission line (F) is 6.5 mm, the cut width of the patch (C) is 0.5 mm, the length of the sensor (P) is 22 mm, the width of the sensor (Q) is 20 mm, and the length of the back layer (E) in the Y direction is 9 mm.

The effective dielectric constant can be defined as
(7)$$\varepsilon_{\mathrm{reff}}=\frac{\varepsilon_{r}+1}{2}+\frac{\varepsilon_{r}-1}{2}{\left[1+12\frac{h}{w}\;\right]}^{-1/2}\;$$
where w is the width of the patch and h is the height of the dielectric substrate.

## 3. Results and Discussion

### 3.1. Metamaterial Unit Cell

The effective parameters, such as effective permittivity and effective permeability, for this proposed metamaterial unit cell are shown in Figure 8. The effective permittivity shows the negative region from 2.16 to 2.36 GHz, 4.04 to 4.47 GHz, and 5.49 to 6.22 GHz. Meanwhile, the negative permeability is exhibited in the negative region for the second resonant frequency from 4.2 to 4.4 GHz. The strong confinement of the applied electromagnetic field on the metal-dielectric-based inclusion depends on the negative effective parameters. The parallel and perpendicular components of the effective permittivity are shown in Figure 9. The parallel component where the wave is propagated in the Y-direction shows the negative region from 2.18 to 2.35 GHz, 4.08 to 4.5 GHz, and 5.45 to 5.6 GHz. Meanwhile, the perpendicular component of the effective permittivity where the wave is propagated in the X-direction exhibits the negative region from 6.11 to 7.57 GHz. The suggested metamaterial shows negative permittivity in three directional propagation modes, which increases the compatibility to be incorporated into the proposed sensor.

For the first resonant frequency in Figure 10a, the electric field is distributed at a high intensity on the horizontal side of the outer metallic strip, in the gap between the metallic strips. For the second resonant frequency, the electric field is distributed at the edge of the outer metallic ring, in the gap between the middle metallic strip and the inner bent metallic strip. At the cutting of the bent metallic layer and the junction of the two bent metallic slots, the electric field accumulated at a high intensity to produce the electric resonance on the metamaterial unit cell. For the third resonant frequency, the confinement of the electric field is exhibited diagonally around the inner bent metallic strip.

The dielectric characteristics of the chosen substrate influence the interaction of the applied electric field, with the dipole moment enhancing polarization for the given electric field. The resonant circular current to the stimulating electric field corresponds to magnetic dipole moments and magnetic resonance couples with the external magnetic field. The magnetic field distribution for the first resonant frequency is shown in Figure 10b, where strong accumulation takes place on the edge of the two metallic square-shaped split rings and the corner edge of the inner metallic bent metallic strip. The excluded area of the electric field distribution exhibited the confinement of the magnetic field on the mutual area between the two square-shaped metallic strips and the edge of the inner bent metallic strip. For the third resonant frequency, the magnetic field is confined strongly on the right side of the inner bent metallic ring.

For this proposed metamaterial design, the current is induced on the outer metallic ring in the clockwise direction where the surface current in Figure 11 is moved in the clockwise direction on the edge of the inner metallic ring. The surface current induced the loop on the edge of the inner metallic ring, where the oppositely directed current with low intensity was induced on the inner bent metallic strip. At the second resonant frequency, the surface current is strongly induced on the outer metallic strip, which follows a clockwise direction, and the anticlockwise-directed current is at the edge of the inner metallic strip of the resonator. The clockwise-directed high-intensity current is distributed at the inner edge of the second metallic strip, and the bend metallic slot of the inner ring exhibits the high intensity of the surface current at the edge region. At the third resonant frequency, the surface current exhibits high intensity at the edge between the outer and inner metallic strips. The strong surface current is distributed on the metallic layer of the inner bent slot in a clockwise direction at the outer edge and an anticlockwise direction at the inner bend edge of the proposed design.

### 3.2. Metamaterial-Incorporated Sensor

A completely integrated microstrip-based sensor with a square-shaped dual-split-ring enclosed bent metamaterial unit cell is integrated at the body center of the design to produce a new resonance peak and boost the sensitivity of the sensor. The square-shaped volumetric sample is utilized, which is attached to the substrate to cover the sensitive region. The proposed metamaterial-based antenna is shown in Figure 12a and has a wider radiation pattern with less directivity, which increases the compatibility of the proposed design as a sensor to enhance the sensitivity around the structure. The electric field distribution in Figure 12b at 6.2 GHz shows the accumulation of the electric field on the gap between the strip patch and the outer split ring of the metamaterial unit cell. For electric field polarization on the proposed metamaterial unit cell, the geometric layout of the metamaterial unit cell inside the microstrip patch, which is deployed in the middle of the patch on the dielectric substrate with a back layer, exhibits the strong accumulation of the electric field component to produce strong scattering resonance. The magnetic field distribution in Figure 12c is exhibited between the vertical edge corner of the microstrip patch antenna and outer-square split-ring resonator. The surface current in Figure 12d is strongly induced on the square-shaped split ring, which is placed in a body-centered position of the microstrip antenna. The current follows a clockwise direction in the right metallic part and an anticlockwise direction in the left part of the proposed sensor. The strong accumulation of the electromagnetic field on the outer metallic arm of the metamaterial unit cell shows a high-sensitivity area for this microwave to characterize the material properties. However, the metamaterial unit cell exhibited three resonant frequencies using the free space method with a two-waveguide-port system. After the insertion of the metamaterial unit cell in the microwave strip patch antenna at the center position of the sensor body. The resonant frequency for the single-transmission line-based microstrip-based sensor is related to the geometric configuration of the microstrip antenna. The geometric layout for this proposed metamaterial-based microstrip sensor exhibited resonance at 6.2 GHz for the reflection coefficient in Figure 13 with a bandwidth from 6.1 to 6.3 GHz. The suggested sensor is directly activated by an incoming EM field with an incorporated transmission line; it has a high level of robustness, integration, non-invasiveness, low manufacturing costs, and higher sensitivity. Moreover, the reflection coefficient of this proposed sensor is validated using CST and HFSS.

### 3.3. Performance Evaluation

The sensitivity of the sensor and resolution are the two most significant characteristics. Sensitivity is a physical property that describes a sensor’s ability to respond to a change in the sample. For sensor performance evaluation, the sensor’s spectrum sensitivity is more relevant, which can be expressed as
(8)$$S=\frac{f_{0}-f_{N\;}}{f_{0}}\;\times \;1000\;(\mathrm{MHz})$$
where 𝑓_0_ represents the center frequency of resonance and 𝑓*_N_* is the new frequency shift of the resonance for different dielectric constants. The smallest variation in the frequency shift that can be detected by the sensor construction is represented as the resolution. The limit of the resonance frequency shift is specified as one-twentieth of the full width at half maximum (FWHM).

The figure of merit (FOM) is used to compare the sensor performance of different bands, where the FOM value can define both the sensitivity and resolution at the same time. The greater the sensor performance, the higher the sensitivity and the smaller the FWHM.
(9)$$\mathrm{FOM}=\frac{S}{\mathrm{FWMH}}$$

A metamaterial-based microstrip antenna with different dielectric materials was simulated to test the performance of the proposed sensor and evaluate the sensitivity of different resonance peaks. The sensitivity with the shift of the reflection coefficient is substantially superior to that of the other substrate for different dielectric constants, as seen in Figure 14. The simulated frequency shift in Table 2 reveals that until the changes in the dielectric environment are big enough, the resonance peaks of SRR-based sensors have a bandwidth of 5.8–6.0, 5.5–5.7, 5.3–5.44, and 5.0–5.26 GHz at 5.9, 5.6, 5.4, and 5.4 GHz, successively, which are insensitive to tiny changes in the dielectric environment.

This finding shows that when diverse dielectric materials are put in the sensor layer and the sensor is operated in the C-band frequency, the suggested metamaterials-based sensor structure can successfully identify them. Moreover, different substrate materials with different dielectric constants of materials can be calculated using a rather accurate model developed in CST based on observed data and applicable to other low-dielectric-constant-based materials. The regression analysis shows the MUTs with different dielectric constant vs. peak frequency, which is linear with curve fitting (as shown in Figure 4), and the actual values fall exactly on the linear curve. The linear regression equation shows a good linear correlation in Figure 15 with different dielectric constants and deviation in the resonance frequency of the reflection coefficient. The relation for sensitivity is y = 37.2x + 15.5, R^2^ = 0.9894, and FOM is y = 125.3x + 4.5, R^2^ = 0.9978.

For the experimental procedure, the microstrip transmission line is connected using an SMA connector and the electromagnetic signal is utilized through a 50Ω coaxial cable from the vector network analyzer, which is shown in Figure 16. The resonant frequencies of the reflection coefficient for the different dielectric materials such as Rogers RT 5880 (*ε* = 2.2) are located at 5.47 GHz, FR-4 (*ε* = 4.3) is located at 5.29 GHz, and air (*ε* = 1) is shown at the resonant frequency at 5.6 GHz. The linear correlation of resonant frequencies for the variation of the dielectric substrate is shown as R^2^ = 0.996.

Interpretation, analysis, and automation of the received data are frequent issues within the approaches, which necessitate highly experienced and specialized professionals. As a result, academics are looking into and using artificial intelligence, namely machine learning algorithms, to address the issues. Researchers have also exhibited different methodologies to increase the accuracy and obtain more parameters to improve measuring procedures. To put the current study in context, the performance of the suggested sensor is compared in Table 3 to that of existing microwave-based sensors in the literature with different parameters such as the unit cell type, substrate material, metamaterial type, and operating frequency range. The performance parameters show different observed values for different ranges of the spectrum coefficient, and we compare the dimensions of the metamaterial unit cell and proposed sensor to that of existing microwave sensors. Moreover, the suggested sensor, with electrically perceptible properties, shows the high intensity of the electric, magnetic, and surface current distributions. The incorporation of the metamaterial unit cell introduces higher-sensitivity areas at the edge of the metallic arm of the unit cell. These numerical and experimental findings show that metamaterial-loaded microstrip-based sensors exhibit the enhanced performance of the sensitivity analysis when the sensor shows frequency deviation with low permittivity values. The suggested metamaterial-based sensor structure can successfully identify them with a good linear correlation for sensitivity and FOM. The dimension ratio for this sensor is 4.04, which is expressed by the compactness of the design from the other mentioned designs in the table. We can conclude that this proposed sensor provides the advantage of size miniaturization, lightweight, low cost, and enhanced sensitivity with good linear correlation of the frequency deviation to increase the durability for commercialization.

## 4. Conclusions

This paper proposes and validates a low-cost and size-miniaturized efficient microwave-based sensor operating at approximately 6.2 GHz. The SRR metamaterial structure can identify various substrates with low permittivity by filling different samples on the small volumetric area, which is the sensitive area. This sensor is directly excited from the coaxial transmission line with a physical connection and has a number of advantages, including low cost, high sensitivity, high stability, reusability, and ease of operation, and has a lot of potential for use in an operating microwave band system. Moreover, the metamaterial unit cell shows negative effective parameters, which ensure the strong accumulation of the electric and magnetic field components to enhance the sensitivity, and the split ring technology increases the compactness of the sensor. This work primarily focused on strengthening the performance of the sensitivity in order to more accurately discriminate between distinct samples as well as improving the resilience of the sensor by addressing the influence of antenna attenuation on the recorded amplitude.

## Figures and Tables

**Figure 1 materials-15-06186-f001:**
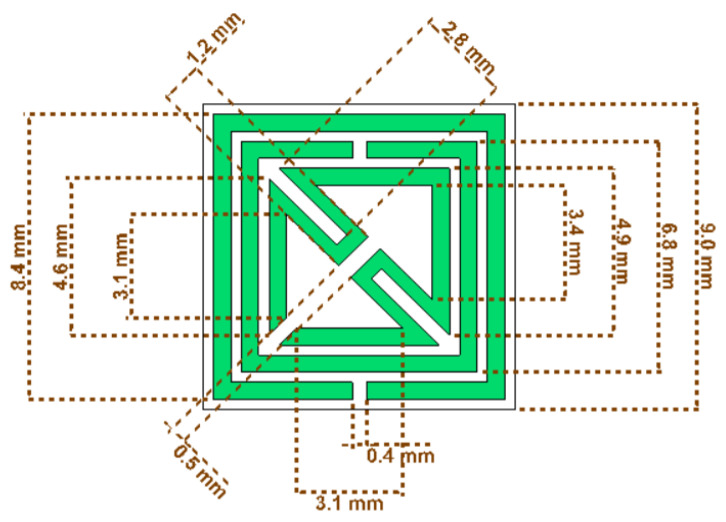
Metamaterial unit cell with specifications of dimensions.

**Figure 2 materials-15-06186-f002:**
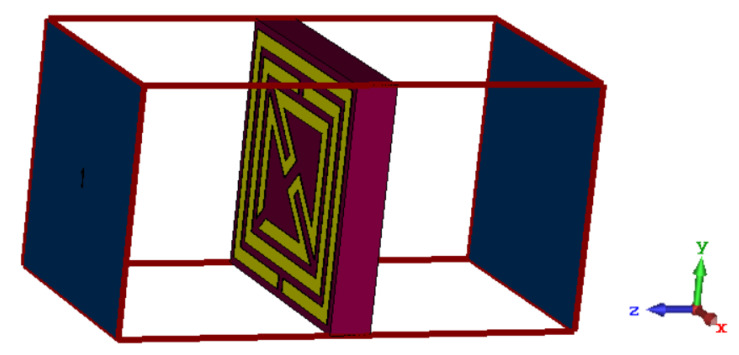
Boundary condition for the interaction of the applied electromagnetic signal. black arrow show the boundary condition of waveguide port excitation.

**Figure 3 materials-15-06186-f003:**
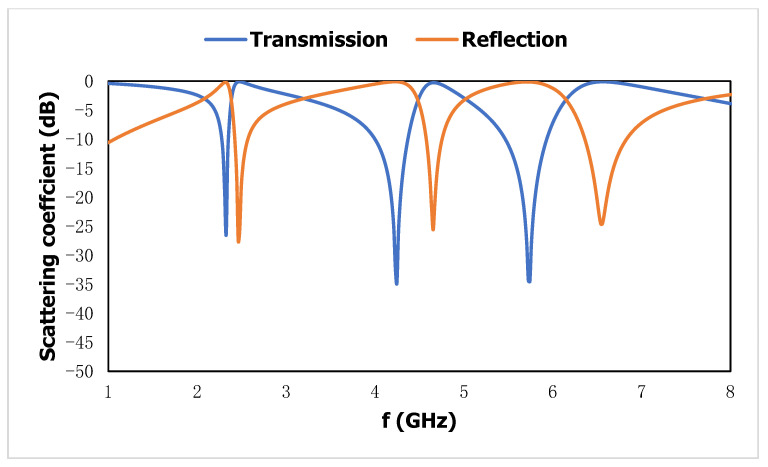
Scattering result of the proposed metamaterial unit cell.

**Figure 4 materials-15-06186-f004:**
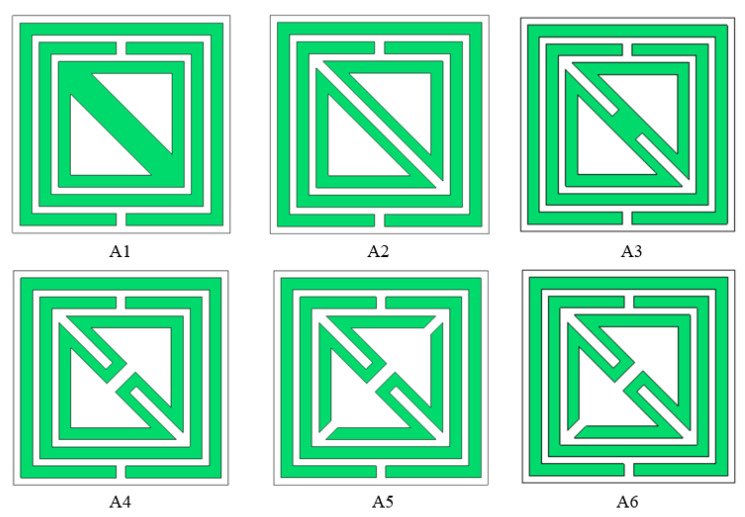
Selection stage for the proposed metamaterial unit cell. (A1) dual square rings with diagonally connected inner ring; (A2) Split the inner ring in diagonally; (A3) inductively coupled in the inner rings; (A4) diagonally split for closed inner ring; (A5) cut the inner ring at the edge of the corner; (A6) connected one edge of the inner ring.

**Figure 5 materials-15-06186-f005:**
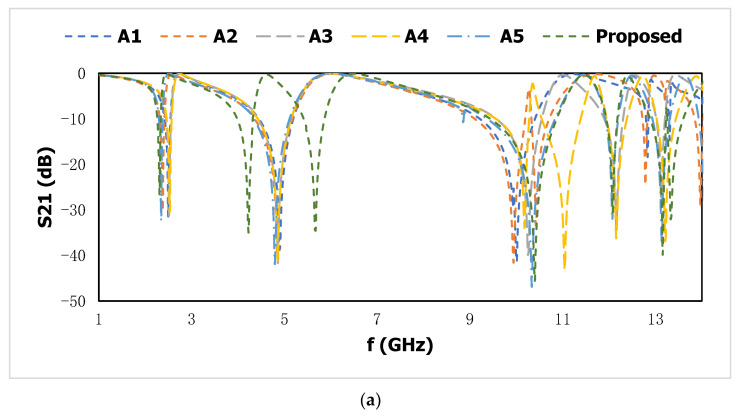
Transmission coefficient (**a**) and reflection coefficient (**b**) for different designs.

**Figure 6 materials-15-06186-f006:**
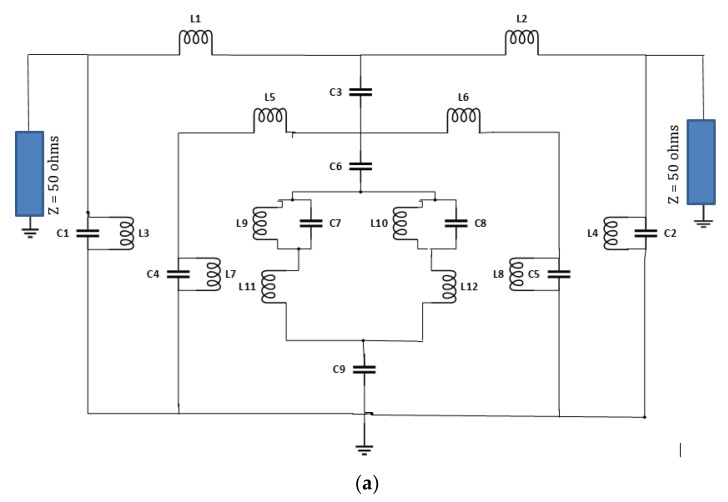
Equivalent circuit model with lumped elements (**a**) and the corresponding results of reflection coefficient (**b**) and transmission coefficient (**c**).

**Figure 7 materials-15-06186-f007:**
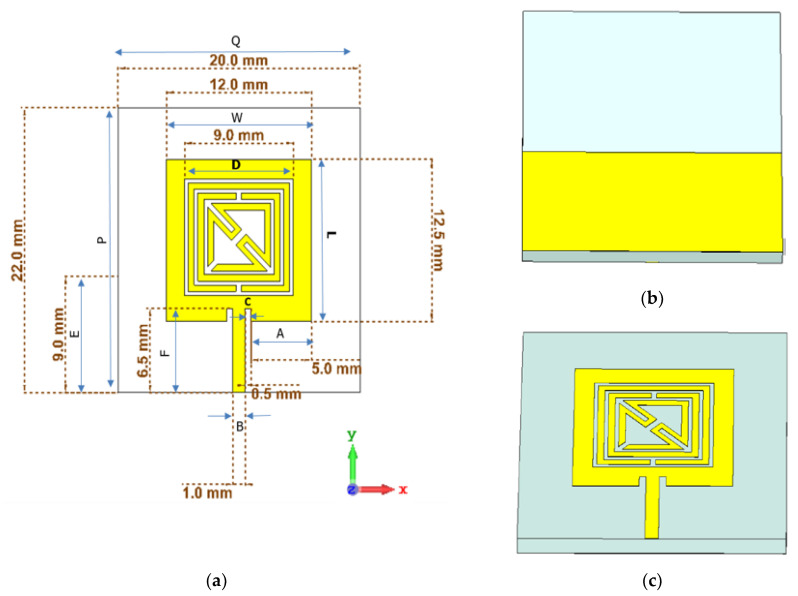
The design configuration with 2D layout (**a**); schematic view of back side (**b**) and front side (**c**) of the proposed metamaterial-based sensor.

**Figure 8 materials-15-06186-f008:**
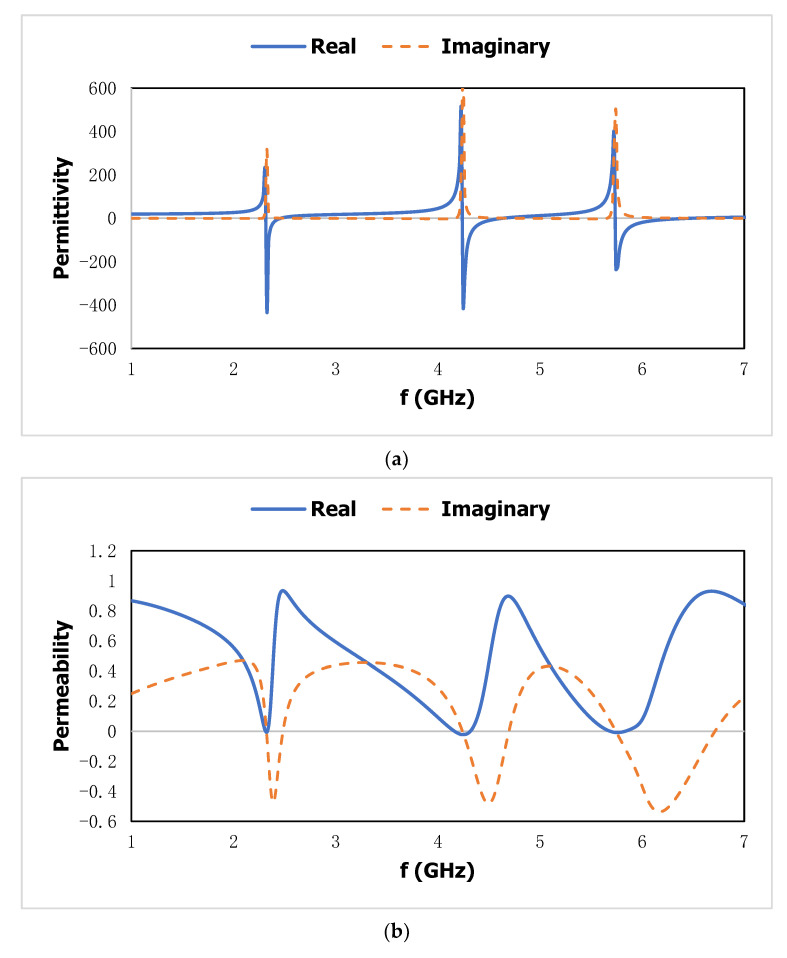
The effective properties of (**a**) effective permittivity; and (**b**) effective permeability of the proposed metamaterial unit cell.

**Figure 9 materials-15-06186-f009:**
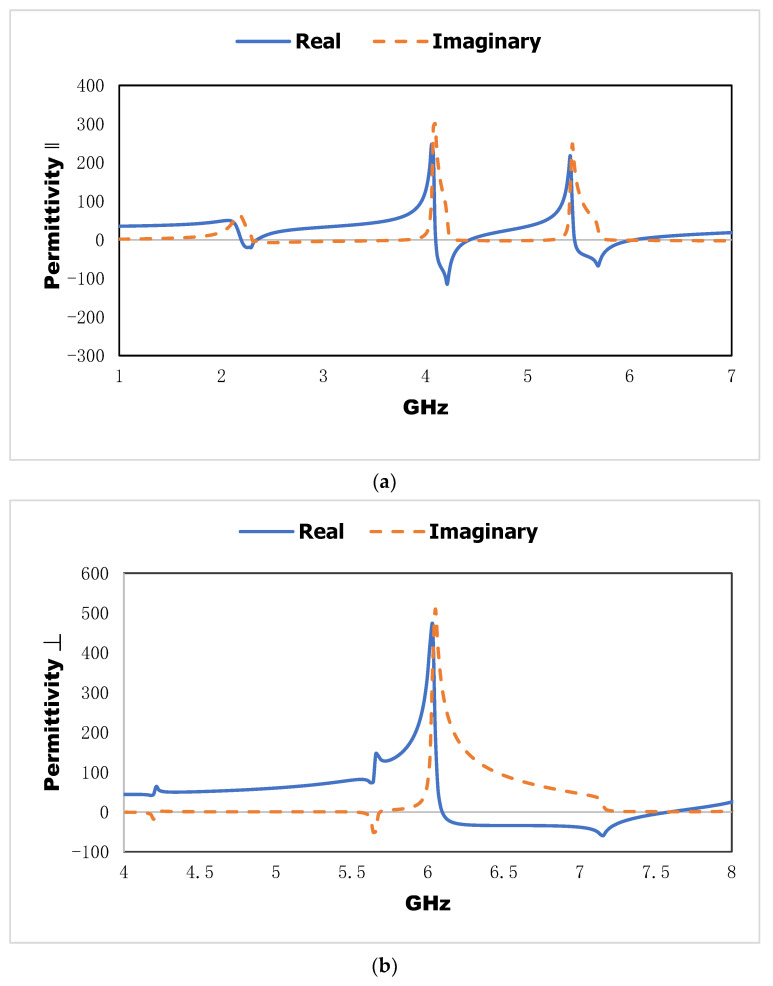
Parallel component (**a**) and perpendicular component (**b**) of the effective permittivity.

**Figure 10 materials-15-06186-f010:**
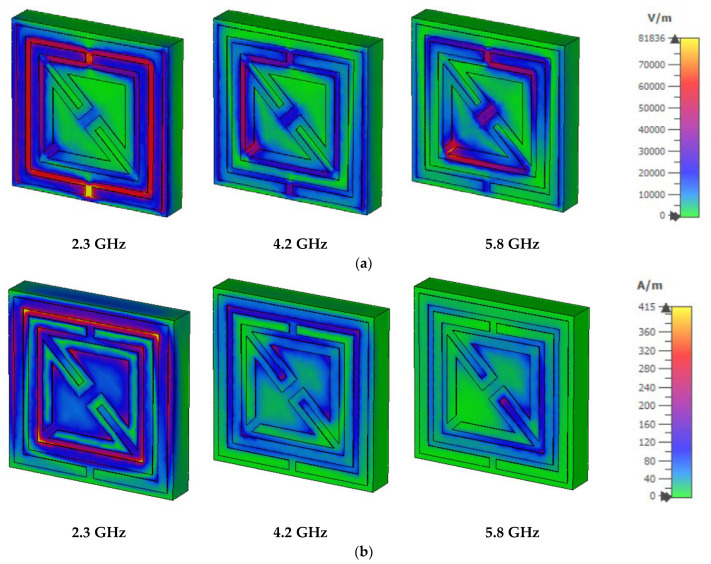
Electric (**a**) and magnetic field (**b**) distribution at 2.3, 4.2, and 5.8 GHz.

**Figure 11 materials-15-06186-f011:**
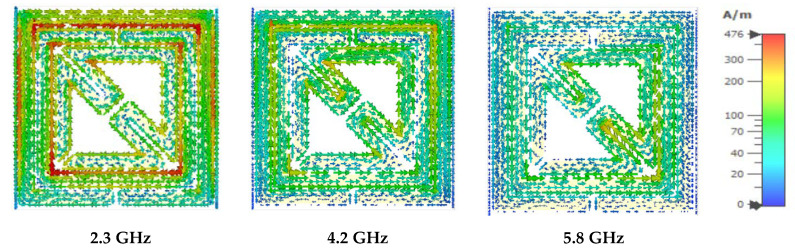
Surface current distribution at 2.3, 4.2, and 5.8 GHz.

**Figure 12 materials-15-06186-f012:**
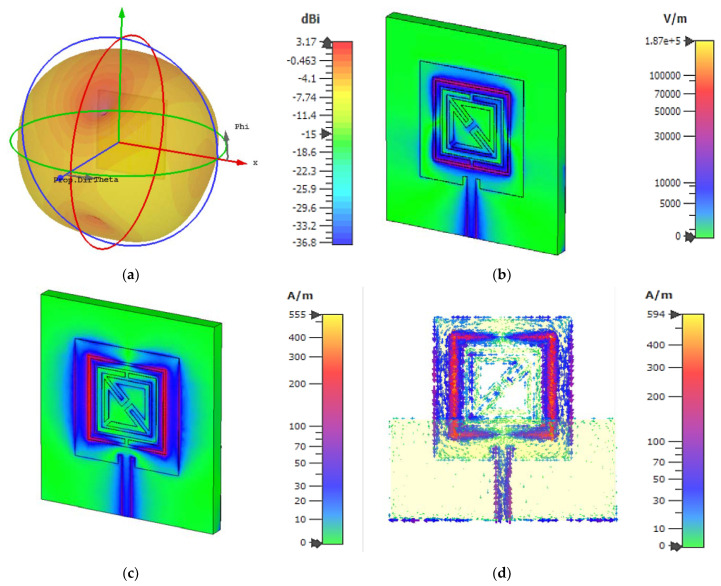
(**a**) Far-field distribution; (**b**) electric field distribution; (**c**) magnetic field distribution; and (**d**) surface current distribution at 6.2 GHz.

**Figure 13 materials-15-06186-f013:**
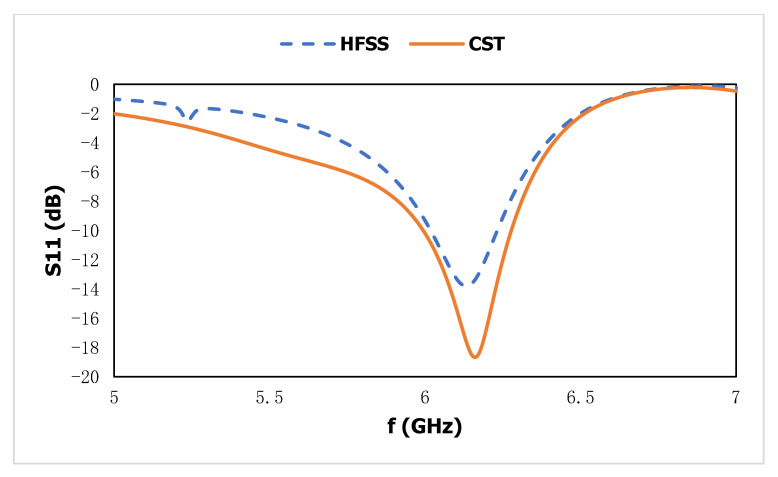
Reflection coefficient of the proposed sensor in CST and HFSS.

**Figure 14 materials-15-06186-f014:**
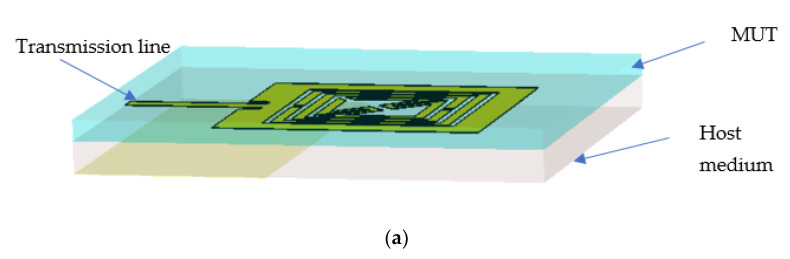
(**a**) 3D view of proposed sensor with MUT and (**b**) corresponding result of reflection coefficient for different MUT.

**Figure 15 materials-15-06186-f015:**
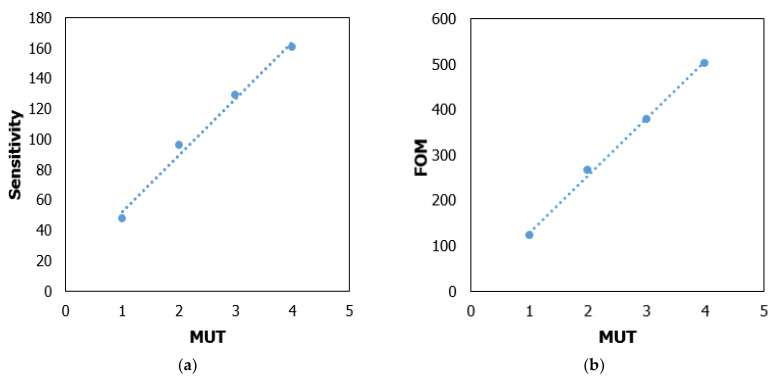
Accuracy measurement with (**a**) sensitivity parameter and (**b**) FOM.

**Figure 16 materials-15-06186-f016:**
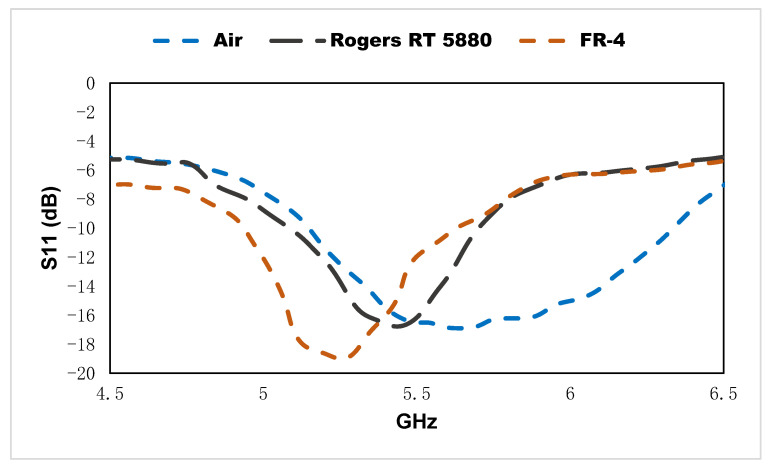
Measured result of the metamaterial-incorporated sensor for different MUTs.

**Table 1 materials-15-06186-t001:** The parametric specifications for the proposed resonator.

Parameter	Width of the Outer Split Ring	Width of the Outer Split Ring	The Gap between the Two Split Rings	The Distance between the Bent Edge	The Width of the Split	The Thickness of the Substrate	The Thickness of the Metallic Layer
**mm**	0.5	0.5	0.3	1.2	0.4	1.524	0.035

**Table 2 materials-15-06186-t002:** Sensing properties for different dielectric constant.

	Dielectric Constant	Bandwidth	Resonance Frequency	FWMH	$S=\frac{f_{0}-f_{N}}{f_{0}}\;\times \;100$	FOM
**Air**	1	6.1–6.3	6.2	0.42	---	---
**MUT-1**	1.5	5.8–6.0	5.9	0.39	48	123
**MUT-2**	2	5.5–5.7	5.6	0.36	96	266
**MUT-3**	2.5	5.3–5.44	5.4	0.34	129	379
**MUT-4**	3	5.0–5.26	5.2	0.32	161	503

**Table 3 materials-15-06186-t003:** Comparative analysis with other established sensors.

	Unit Cell Dimension (mm)	Sensor Dimension (mm)	Substrate	Type	Operating Band (GHz)	Dimension Ratio (λ/D)	Remarks
[18]	183.5 × 183.5	183.5 × 183.5	ISOLA IS680	SRR	1–3	0.57	Concrete sample detection
[19]	3 × 3	30 × 25	FR-4	CSRR based antenna	1–4	3.7	detection of the different substrate
[21]	10.16 × 22.86	10.16 × 22.86	FR-4	Rectangular SRR	8–12	1.31	Liquid oil detection
[22]	10 × 22	10 × 22	FR-4	Metamaterial Absorber	8–12	1.38	Liquid chemical detection
[23]	7 × 7	24 × 60	FR-4	SRR based antenna	2–6.5	1.56	Different substrate detection
Proposed	9 × 9	20 × 22	Rogers RT 6002	SRR based antenna	4–8	4.03	Detection of the substrate material

## Data Availability

All data are available within the manuscript.

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
