# Peer review of "Body-Centered Double-Square Split-Ring Enclosed Nested Meander-Line-Shaped Metamaterial-Loaded Microstrip-Based Resonator for Sensing Applications"

_materials, 2022, doi:10.3390/ma15186186_

Round 1

Reviewer 1 Report

The paper presented a design of a metamaterial-based sensor implemented using meandered split ring resonators. The manuscript is fair in general; find below my comments:

11- In general, paragraphs throughout the manuscript are not connected well. Is there no transition between the reported results?  

22- In section 2, lines 130 and 131, the reported transmission coefficient resonance frequency at 6 GHz is inconsistent with the BW corresponding for this resonance from 5.32 to 5.82 GHz?

33- What are the properties of the incident field used to characterize the reflection and transmission coefficient of the unit cell (E-field direction - angle of incident)?

44- Can the authors elaborate more on the purpose of the optimization study reported on pages 5 and 6?

55- The idea behind the equivalent circuit model is not clear. In addition to the equations reported on page 8? Please explain and justify the purpose of the equivalent circuit and the Eqs.

66- Based on the proposed design sensor, is the main contribution behind the study unclear? Does it have a better sensitivity or compact size than the other literature sensors?

77- Table 3 and the discussion do not show any benefit or advantage of the proposed sensor compared with the reported one; for example, in Table 3?

Author Response

As attached.

Reviewer 2 Report

In this research work, the current effort involves the design of a novel metamaterial based sensor that can be used to detect substances within the frequency range of 4 to 8 102 GHz. I think paper is interesting, I would propose some changes as follows:

1.      It would be desirable to confirm the obtained results by comparing them with some experimental outputs.

2.      Authors should conclude based on Fig. 7 what type of the metamaterial they are investigating. Moreover, it would be of particular interest to plot parallel and perpendicular permittivity components.

3.      Authors are missing some recent articles in the field such as Analytic solution to field distribution in two-dimensional inhomogeneous waveguides.

4.      Authors should stress novelty of their work in comparison with others.

Author Response

As attached.

Round 2

Reviewer 1 Report

The authors addressed my comments and made the necessary changes to the manuscript.